# Efficacy and Safety of Non-Insulin Antidiabetic Drugs in Cats: A Systematic Review

**DOI:** 10.3390/ani15172561

**Published:** 2025-08-31

**Authors:** Félix Romero-Vélez, Juan Rejas, Rafael Ruiz de Gopegui

**Affiliations:** 1Hospital Clínic Veterinari, Universitat Autònoma de Barcelona, 08193 Bellaterra, Spain; 2Department of Animal Medicine, Surgery and Anatomy, Universidad de León, 24071 León, Spain; juan.rejas@unileon.es; 3Department of Animal Medicine and Surgery, Universitat Autònoma de Barcelona, 08193 Bellaterra, Spain

**Keywords:** feline diabetes mellitus, SGLT2 inhibitors, velagliflozin, bexagliflozin, glipizide, exenatide, evidence-based medicine

## Abstract

Feline diabetes mellitus is a common disease typically requiring daily insulin injections, which can be challenging for cats and their owners. This has prompted interest in non-insulin antidiabetic drugs (NIADs). We conducted the first comprehensive systematic review to gather and critically evaluate all scientific evidence on the use of these NIADs in cats. Our review of 20 studies found that older oral drugs like glipizide have limited efficacy and may even be harmful to the pancreas. In contrast, a new class of once-daily oral drugs, the SGLT2 inhibitors (bexagliflozin and velagliflozin), are highly effective, often working as well as insulin. However, their main risk is not low blood sugar (hypoglycemia) but a different serious side effect called euglycemic diabetic ketoacidosis (eDKA), which requires careful monitoring. This review shows that veterinarians and cat owners now have a viable oral alternative to insulin, but the choice of treatment involves a new risk–benefit discussion centered on owner education and vigilance for signs of eDKA.

## 1. Introduction

Feline diabetes mellitus (FDM) is one of the most common endocrine disorders in domestic cats [1,2,3]. It is characterized by impaired glucose homeostasis due to progressive pancreatic beta-cell dysfunction, peripheral insulin resistance, or a combination of both mechanisms [1,4]. Obesity, acromegaly, dental and kidney diseases, systemic infection, pancreatitis, pregnancy/diestrus, medications (e.g., corticosteroids, progestins, cyclosporine), advanced age, and specific breed predispositions such as in Burmese cats are considered key risk factors contributing to the onset and progression of FDM [4,5].

The clinical management of FDM often poses challenges due to stress-induced hyperglycemia, variability in treatment response, and the need for owner compliance with daily insulin administration. Furthermore, achieving adequate glycemic control with insulin can be difficult, with a significant proportion of cats failing to achieve diabetic remission, often due to underlying insulin-resistant conditions such as hypersomatotropism [6]. While the mainstay of treatment for clinical FDM is insulin along with dietary modification [4,7], interest has grown in using non-insulin antidiabetic drugs (NIADs), especially in scenarios where insulin therapy is not feasible. Several classes of NIADs have been evaluated in feline patients, including sulfonylureas (e.g., glipizide) [8,9,10,11,12,13], biguanides (e.g., metformin) [5,14,15], alpha-glucosidase inhibitors (e.g., acarbose) [16,17,18], glucagon-like peptide-1 (GLP-1) receptor agonists (e.g., exenatide) [2,3,19,20,21,22,23], chromium [24,25], thiazolidinediones (e.g., darglitazone) [25,26,27,28], and, more recently, sodium-glucose cotransporter 2 (SGLT2) inhibitors (e.g., velagliflozin) [1,29,30,31,32,33].

These agents function through distinct mechanisms of action: sulfonylureas, such as glipizide, primarily act by stimulating insulin secretion from pancreatic ß-cells [11]; biguanides, represented by metformin, mainly reduce hepatic gluconeogenesis [14]; alpha-glucosidase inhibitors, like acarbose, function by delaying carbohydrate digestion and glucose absorption from the intestine [17]; thiazolidinediones (pioglitazone and darglitazone) are insulin sensitizers that act as peroxisome proliferator-activated receptor gamma (PPARγ) agonists [25,28]; GLP-1 receptor agonists, such as exenatide, are incretin mimetics that enhance glucose-dependent insulin secretion and suppress glucagon [2,19]; more recently, SGLT2 inhibitors (bexagliflozin and velagliflozin) have emerged, lowering blood glucose (BG) by inhibiting its reabsorption in the renal tubules, thereby promoting urinary glucose excretion [1]; and chromium has been evaluated for its role as a trace mineral cofactor for insulin function [25]. The distinct mechanisms of action for these drug classes are summarized in Figure 1.

Despite increasing interest in these alternative treatments, the evidence regarding the efficacy and safety of NIADs in cats with FDM remains limited and fragmented, often characterized by heterogeneous methodologies and outcomes, and frequently based on small cohorts or single case reports. While some drugs have demonstrated potential benefits in improving glycemic control or increasing the likelihood of diabetic remission [30], others have shown limited efficacy [34] or notable adverse effects [1,20].

To date, there is no comprehensive synthesis of the scattered clinical evidence addressing the role of NIADs in FDM. Such a synthesis is needed to support evidence-based decision-making in feline diabetic care.

This review aims to describe the dosing regimens, therapeutic efficacy, and adverse effects associated with NIADs; assess the role of these drugs as primary or adjunctive treatments in the management of FDM; identify their clinical indications and potential limitations to support their use in cats with FDM or those at risk of developing it; and, finally, highlight current gaps in the literature and suggest directions for future research.

## 2. Materials and Methods

### 2.1. Protocol

A systematic review following the Preferred Reporting Items for Systematic Reviews and Meta-Analyses (PRISMA) guidelines was conducted [35]. The protocol for this systematic review was established a priori but was not eligible for registration in the PROSPERO database, as the review had already progressed to the data extraction phase at the time of the submission attempt.

### 2.2. Eligibility Criteria and Study Selection

A systematic literature search was conducted in three databases: Google Scholar, PubMed, and Scopus (Table 1). It is important to note that the search strategy was designed to be comprehensive and captured all NIADs classes across all databases. The specific interventions listed in the PICO A and PICO B criteria therefore reflect the drug classes for which eligible primary studies were ultimately identified for each respective population. No restrictions were applied regarding publication dates or language. All retrieved records were imported into Mendeley Reference Manager (Version 2.134.0), where duplicates were first removed automatically and then manually verified.

The study selection process followed a two-stage screening protocol performed independently by two reviewers (Rafael Ruiz de Gopegui and Juan Rejas). In the first stage, both reviewers independently screened the titles and abstracts of all unique records to identify potentially relevant articles. In the second stage, the full texts of these articles were retrieved and assessed independently by the same two reviewers against the predefined inclusion and exclusion criteria (Table 1). Any discrepancies at either stage were resolved through discussion and consensus. If an agreement could not be reached, a third reviewer (Félix Romero-Vélez) was consulted to make the final decision. The entire selection process is documented in the PRISMA flow diagram (Figure 2).

### 2.3. Risk of Bias Assessment

The risk of bias for each included study was assessed using the appropriate tool based on the study design. The Cochrane Risk of Bias 2 (RoB 2) tool was used for randomized controlled trials (RCTs), and the Risk Of Bias In Non-randomised Studies of Interventions (ROBINS-I) tool was used for non-randomized studies. The assessment was performed independently by two reviewers, with disagreements resolved by consensus. Comprehensive risk of bias judgments for individual studies is presented with detailed justifications in Appendix A.

### 2.4. Declaration on the Use of Artificial Intelligence

During the preparation of this manuscript, the authors used Gemini 1.5 Pro for the purposes of refining the manuscript’s formatting to adhere to journal-specific guidelines, ensuring compliance with PRISMA 2020 reporting standards, clarifying ethical and submission requirements, and correcting textual errors to enhance clarity and native English fluency. The authors have reviewed and edited the output and take full responsibility for the content of this publication.

## 3. Results

### 3.1. Study Selection

The initial search from the three databases yielded 240 articles. After removing duplicates, 150 records were screened. A total of 148 full-text articles were assessed for eligibility, of which 21 reports, describing 20 unique studies, met the inclusion criteria. One report [21] was identified as a secondary analysis of a previously included trial [19] and data from both reports were therefore synthesized as a single study. The remaining reports were excluded for several reasons, primarily, being abstract-only publications, not meeting the PICO criteria (e.g., studies in healthy cats without risk factors), focusing exclusively on insulin therapy, or being review articles without primary data. The final selection process is illustrated in the PRISMA flow diagram (Figure 2).

### 3.2. Studies in Cats with Diabetes Mellitus

#### 3.2.1. Characteristics of Included Studies

A total of 10 studies reporting on the treatment of cats with naturally acquired FDM were included (Table 2). The publication dates ranged from 1993 to 2024. The study designs were heterogeneous: two were RCTs, two were non-RCTs, and six were uncontrolled prospective trials (case series). The studies were conducted primarily in North America and Europe and included a combined total of 605 cats. The reported breed composition of the cats included in these trials is summarized in Table 3.

#### 3.2.2. Synthesis of Results

##### Sulfonylureas (Glipizide)

Two early, prospective uncontrolled trials investigated the efficacy of the sulfonylurea glipizide. Nelson et al. [11] treated 20 diabetic cats with glipizide (5 mg/cat BID) for 12 weeks, reporting that 25% (5/20) achieved good glycemic control, 40% (8/20) had a partial response, and 35% (7/20) failed to respond to treatment. Vomiting, hypoglycemia, and elevated liver enzymes were reported as adverse effects.

In a long-term trial (50 weeks) Feldman et al. [10] administered glipizide to 50 newly diagnosed diabetic cats. A high rate of treatment failure was observed, with 56% (28/50) of cats requiring a switch to insulin. Among the other cats, 26% (13/50) were considered sustained responders with clinical or metabolic improvement, and 12% (6/50) experienced diabetic remission after episodes of hypoglycemia. Reported adverse effects included transient anorexia, vomiting, and icterus.

##### Alpha-Glucosidase Inhibitors (Acarbose)

One non-RTC investigated the alpha-glucosidase inhibitor, acarbose, as an adjunct therapy. Mazzaferro et al. [17] studied 24 diabetic cats with a history of obesity, comparing a group treated with acarbose, a low-carbohydrate diet, and insulin to a control group receiving only the diet and insulin. Although a high proportion of cats receiving acarbose (61%, 11/18) were able to discontinue insulin, the authors concluded that this result was likely attributable to the diet itself, with acarbose having a minimal clinical effect. No adverse effects related to the drug were reported.

##### Biguanides (Metformin)

The biguanide metformin was assessed as monotherapy in a single, small, uncontrolled prospective trial. Nelson et al. [14] administered metformin to five cats with newly diagnosed diabetes. Efficacy was found to be low, with only one cat (20%) showing a sustained improvement in glycemic control. Three cats (60%) failed to respond to treatment and were subsequently transitioned to insulin, while one cat died unexpectedly during the study period. Lethargy, inappetence, and vomiting were noted as adverse effects in healthy cats during the safety phase of the trial.

##### Glucagon-like Peptide-1 (GLP-1) Receptor Agonists (Exenatide)

The GLP-1 receptor agonist exenatide was assessed as an adjunct therapy to insulin in two clinical trials. The first was a non-RTC in 30 newly diagnosed diabetic cats, whose primary outcomes were reported by Riederer et al. [19]. While a higher numerical remission rate was observed in the exenatide group compared to placebo (40% vs. 20%), this difference was not statistically significant. A secondary analysis of this same cohort by Krämer et al. [21] found that exenatide significantly reduced glycemic variability. The second clinical trial was a double-blinded, randomized crossover study in eight diabetic cats by Scuderi et al. [2]. This study found that short-acting exenatide significantly lowered the required daily insulin dose and promoted weight loss compared to placebo. Two cats (25%) in this trial achieved remission while on exenatide, versus none on placebo. Reported adverse effects across both studies were primarily mild and gastrointestinal in nature, including decreased appetite and vomiting.

##### SGLT2 Inhibitors (Bexagliflozin and Velagliflozin)

Bexagliflozin was assessed in two prospective uncontrolled trials. A small pilot study by Benedict et al. [31] in five poorly regulated diabetic cats found that adding bexagliflozin significantly reduced insulin dose requirements. A subsequent large field study by Hadd et al. [32] evaluated bexagliflozin as monotherapy in 84 newly diagnosed diabetic cats, reporting a treatment success rate of 84% at day 56.

Velagliflozin was the subject of two large, pivotal trials with different designs. A prospective, baseline-controlled trial in 252 cats by Behrend et al. [1] found that 81% of remaining cats achieved glycemic control at day 180. In parallel, a randomized controlled trial by Niessen et al. [30] compared velagliflozin to insulin in 127 cats and demonstrated non-inferiority for achieving treatment success.

A consistent safety concern across the SGLT2 inhibitor studies was the risk of eDKA, which was reported as a serious adverse event in both bexagliflozin and velagliflozin trials [1,30,32]. Other commonly reported adverse effects included gastrointestinal signs such as diarrhea.

### 3.3. Studies in At-Risk or Experimental Feline Models

#### 3.3.1. Characteristics of Included Studies

A total of 10 studies investigating the effects of NIADs in at-risk or experimental feline models were included (Table 4). The publication dates ranged from 1999 to 2022. The included study designs were predominantly RCTs (n = 8), with the remaining two being non-RCTs. The studies were conducted primarily in North America and Europe, with one study from South America, and included a combined total of 138 cats.

#### 3.3.2. Synthesis of Results

##### Effects on Insulin Sensitivity in Obese Cats

Several studies evaluated the effects of different NIADs on insulin sensitivity in healthy, obese cats, which serve as a natural model for insulin resistance. Two studies assessed the effects of thiazolidinediones (TZDs). In a randomized, placebo-controlled trial, Hoenig & Ferguson (2003) [26] found that darglitazone significantly improved insulin sensitivity and lipid metabolism. Similarly, a crossover trial by Clark et al. [28] reported that a 3 mg/kg dose of pioglitazone also significantly improved insulin sensitivity. The SGLT2 inhibitor velagliflozin was also assessed in this model by Hoenig et al. [29] in a non-randomized, placebo-controlled clinical trial; while it did not significantly change the rate of glucose clearance, the cats required less insulin to clear the same amount of glucose, suggesting an improvement in insulin sensitivity. In contrast, the GLP-1 agonist exenatide, evaluated in a crossover trial by Hoelmkjaer et al. [38], did not significantly improve insulin sensitivity, although it did induce weight loss. Finally, a non-randomized, placebo-controlled clinical trial by Cohn et al. [25] also found that dietary supplementation did not affect glucose tolerance. In these studies, on healthy obese cats, all evaluated drugs were generally reported to be well-tolerated.

##### Effects in Healthy, Non-Obese, or Mixed-Weight Cats

Three studies investigated the effects of NIADs in healthy cats that were either lean or of mixed body condition. Two studies evaluated GLP-1 receptor agonists with differing results. In a randomized, placebo-controlled, crossover trial, Hall et al. [22] found that a single injection of liraglutide successfully increased glucose-stimulated insulin secretion and suppressed glucagon in a group of lean and overweight cats, with no adverse effects reported. In contrast, a similar randomized, placebo-controlled, crossover trial by Rudinsky et al. [37] reported that a long-acting exenatide formulation did not significantly alter glucose or insulin dynamics, though it did cause a significant reduction in food intake and a high incidence of vomiting. Finally, a non-randomized, placebo-controlled clinical trial by Cohn et al. [25] assessed dietary chromium supplementation and found no effect on glucose tolerance in healthy non-obese cats; no adverse health effects were observed in this group.

##### Effects on Specific Pathophysiological Models

Two studies included in this systematic review evaluated the effects of NIADs on specific, induced pathophysiological states relevant to FDM. In a randomized controlled trial using an experimental model, Hoenig et al. [9] investigated the formation of islet amyloid after artificially inducing FDM in healthy cats. In this model, all cats treated with the insulin secretagogue glipizide developed islet amyloid deposits over 18 months, compared to only one of four cats treated with insulin, suggesting that chronic ß-cell stimulation may accelerate this pathological process. No clinical adverse effects were reported during the trial. In a different model, a randomized clinical trial by Leal et al. [5] investigated the potential protective effect of metformin against corticosteroid-induced insulin resistance. The study concluded that neither metformin nor a specialized diet was effective in preventing the insulin resistance caused by a single methylprednisolone acetate injection, and no side effects associated with the interventions were reported by the owners.

##### Pharmacokinetic Studies

Finally, one study focused exclusively on the pharmacokinetic properties of a thiazolidinedione in healthy cats. The randomized crossover trial by Clark et al. [36] investigated the oral bioavailability of pioglitazone in both lean and obese cats. The study found that bioavailability was high (mean 86%) and that there were no significant differences in key pharmacokinetic parameters between the two groups. No adverse effects were reported during the trial.

## 4. Discussion

### 4.1. Principal Findings and Overview of the Evidence

In line with the objectives established for this review, the following sections will discuss the efficacy, safety, and limitations according the available evidence for NIADs in cats. The principal finding is that while older agents such as sulfonylureas and biguanides are supported by limited and generally low-quality evidence, newer drug classes, particularly the SGLT2 inhibitors, have shown considerable efficacy in recent, larger clinical trials. However, this body of evidence is marked by significant heterogeneity in study designs. The evidence base consists of a small number of randomized controlled trials alongside a majority of uncontrolled prospective studies, many of which are limited by small sample sizes and a lack of statistical power. Furthermore, confounding factors such as diet and disease chronicity were often not controlled, which, combined with the observational nature of many trials, limits the ability to establish clear causality. Accordingly, the following discussion will interpret these findings for each population group sequentially

### 4.2. Interpretation of Evidence from Clinical Treatment Studies

#### 4.2.1. Clinical Efficacy: Traditional vs. Newer Agents

The evidence for NIADs in diabetic cats reveals a marked difference in reported efficacy and in the quality of evidence between traditional and newer drug classes. The evidence supporting older agents like sulfonylureas, alpha-glucosidase inhibitors, and biguanides is not only limited but is also derived almost exclusively from small, single-center, uncontrolled studies from over two decades ago, which carry a high risk of bias. Efficacy in these early trials was often modest, with high rates of treatment failure for glipizide [10,11], minimal clinical effect attributed to acarbose [17], and low efficacy for metformin [14]. Critically, the definition of “treatment success” in these studies often focused on improvements in surrogate metabolic parameters. This highlights an important limitation in the evidence base: a statistically significant change in a parameter like fructosamine does not always translate to a clinically meaningful outcome for the patient, such as durable diabetic remission or improved quality of life.

In contrast, the evidence for newer drug classes, particularly the SGLT2 inhibitors, is supported by larger, more recent, multicenter clinical trials, including active-controlled RCTs [1,30,32]. These studies report high rates of treatment success, which is often more robustly defined as a composite of both glycemic control and owner-assessed improvement in clinical signs. The superior efficacy of SGLT2 inhibitors may be partly attributable to their insulin-independent mechanism of action—promoting urinary glucose excretion—which is effective even in cats with significant ß-cell dysfunction. This is a key difference from agents like glipizide, whose reliance on stimulating potentially exhausted ß-cells may explain their limited long-term success. Similarly, the GLP-1 agonist exenatide, when used as an adjunct to insulin in a high-quality RCT, demonstrated clinically relevant benefits such as reduced insulin requirements and diabetic remission in some cats [2].

#### 4.2.2. Confounding Factors and Effect Modifiers in Clinical Trials

##### Diet

A critical confounding factor across the reviewed clinical trials is the significant, and often uncontrolled, influence of diet. This is best exemplified by the study on acarbose by Mazzaferro et al. [17], where the authors concluded that the high rate of insulin discontinuation observed was likely attributable to the potent effects of the concurrent low-carbohydrate diet, masking any potential minor contribution from acarbose. This highlights that diet is a potent effect modifier, and the efficacy of NIADs cannot be fully interpreted without considering the dietary context.

Furthermore, the mechanism of action of certain drug classes is intrinsically linked to diet. The effect of alpha-glucosidase inhibitors, which delay carbohydrate absorption, is logically dependent on the presence of carbohydrates in the diet. Similarly, the magnitude of glucosuria induced by SGLT2 inhibitors is proportional to the filtered glucose load, which is influenced by dietary carbohydrate content.

Finally, the well-documented ability of low-carbohydrate diets alone to induce diabetic remission in cats complicates the interpretation of remission rates reported in several trials [4]. The lack of dietary standardization across most of the included trials therefore represents a major limitation, making it difficult to isolate the true pharmacological effect of the evaluated agents and to compare outcomes between studies. Furthermore, in field studies involving client-owned cats, owner compliance with any dietary recommendations is an additional significant and unmeasured variable that can introduce further variability in outcomes.

##### Age and Chronicity of the Disease

While the age of cats across the clinical trials was relatively consistent (typically middle-aged to senior), a more significant source of heterogeneity was the chronicity of the diabetic state at enrollment. Several key studies focused exclusively on newly diagnosed, treatment-naïve cats [10,32], whereas others included cats with established, poorly regulated FDM that were already receiving insulin [2,31]. The largest trials included a mixed population of both newly diagnosed and previously treated individuals [1,30].

This distinction is clinically crucial. It is plausible that cats with newly diagnosed FDM may retain more residual pancreatic ß-cell function compared to those with long-standing disease. Consequently, therapies that depend on stimulating endogenous insulin secretion, such as sulfonylureas, might be expected to have greater efficacy in this treatment-naïve population. Conversely, the high failure rate of glipizide even in newly diagnosed cats [10] could suggest that significant ß-cell exhaustion is already present at the time of diagnosis in many patients. This limitation would theoretically be less relevant for therapies with an insulin-independent mechanism, such as SGLT2 inhibitors, potentially explaining their high efficacy rates in the large, mixed-population trials [1,30]. The lack of stratification of results based on disease duration in these trials, however, makes it difficult to confirm this hypothesis.

##### Breed and Genetic Factors

Another potential modifying factor is the genetic background of the cats included in the trials. A review of the included studies reveals that the vast majority of participants were described as Domestic Shorthair, Domestic Longhair, or mixed-breed cats [10,11,19,31]. This is particularly relevant for breeds with a known genetic predisposition for FDM, such as the Burmese cat, in which the underlying pathophysiology has been linked to both reduced insulin sensitivity and altered ß-cell function [4]. Consequently, it is plausible that a breed whose diabetic phenotype is driven primarily by secretory failure might respond poorly to an insulin secretagogue like glipizide but could still be a candidate for an insulin-independent therapy such as an SGLT2 inhibitor. Therefore, it remains unknown whether the efficacy and safety profiles of NIADs observed in the general cat population can be extrapolated to these at-risk breeds, representing a gap in the current evidence.

##### Comorbidities and Concomitant Medications

The external validity—or generalizability—of the findings from the reviewed trials is significantly impacted by the management of comorbidities. Most pivotal trials implemented strict exclusion criteria, effectively selecting for a population of diabetic cats that were otherwise healthy and often not receiving other medications [1,10,32]. While this approach reduces confounding, it creates a substantial gap between the trial populations and the typical older, multi-morbid feline diabetic patient. This is particularly relevant for common comorbidities that cause severe insulin resistance and subsequent treatment failure, such as hypersomatotropism (acromegaly), which may affect up to 25% of diabetic cats and is a leading cause of poor glycemic control [6,39].

This limitation is of paramount importance for the SGLT2 inhibitors. The risk for their most serious adverse effect, eDKA, is known to be precipitated by concurrent illness or periods of anorexia [1,32]. Therefore, the exclusion of cats with common subclinical conditions like chronic pancreatitis from pivotal trials means the true incidence and risk of DKA in a general practice population may be underestimated. Furthermore, the lack of data on cats receiving concurrent medications for other diseases leaves a critical knowledge gap regarding potential drug–drug interactions.

#### 4.2.3. Safety Profile and Adverse Events

Beyond efficacy, the safety profile is a critical determinant of the clinical utility of any NIAD, and this systematic review identified distinct risk profiles for each drug class. For traditional agents, glipizide was associated with a risk of hypoglycemia and hepatotoxicity [10,11]. Metformin was linked to poor tolerability and gastrointestinal signs, and while one cat died unexpectedly during its trial, a definitive cause was not established, making a direct link to the drug speculative [14]. The GLP-1 agonists were generally well-tolerated, with the most common adverse effects being mild and transient gastrointestinal signs [2,10].

The most significant safety finding from this review relates to the SGLT2 inhibitors. It is crucial to differentiate the nature of their primary adverse event from that of older drugs. While hypoglycemia is a predictable extension of the pharmacological effect of a secretagogue, eDKA, consistently reported in SGLT2 trials [1,30,32], is a complex metabolic complication that represents a novel clinical challenge. It is also important to note that the reported incidence rates of eDKA, while clinically significant, are derived from samples with limited precision. For instance, the combined incidence from the largest trials is approximately 7% (95% CI calculated by authors: 4.5% to 10.5%), a range that highlights that the true population risk remains uncertain [1,30,32].

This distinction has direct implications for monitoring, shifting the focus from solely preventing hypoglycemia to also educating owners on diligent observation for subtle signs of illness (eg, anorexia, lethargy) and regular screening for ketonuria [7]. Ultimately, this highlights a fundamental risk–benefit trade-off: the high efficacy reported for SGLT2 inhibitors must be weighed against the significant risk of eDKA, a life-threatening complication that requires a high degree of owner commitment and education, posing a significant practical challenge for both veterinarians and cat owners in a clinical setting.

#### 4.2.4. Strengths and Limitations of the Evidence

A critical appraisal of the evidence for the clinical treatment of diabetic cats reveals significant methodological limitations across the majority of the included studies. The evidence base is dominated by studies with a moderate to serious risk of bias, primarily driven by the lack of a concurrent control group in many key trials, including the largest field studies [1,10,11,14,31,32]. This is particularly important when interpreting the high success rates (>80%) reported for SGLT2 inhibitors, as these findings originate from uncontrolled trials where susceptibility to confounding and placebo effects is high [1,32].

This creates a dichotomy in the evidence. On the one hand, large field studies offer high external validity, as they reflect a diverse, real-world population. On the other hand, the two randomized controlled trials [2,30] provide higher internal validity, giving more confidence in their specific findings. The double-blinded, placebo-controlled, crossover design of Scuderi et al. [2], for instance, represents the gold standard for minimizing bias. Therefore, while newer agents appear highly effective, these promising findings must be interpreted with considerable caution. The more modest results from the higher-quality RCTs may represent a more realistic expectation of efficacy.

### 4.3. Interpretation of Evidence from At-Risk and Experimental Models

#### 4.3.1. Effects on Insulin Sensitivity and Metabolism

The studies conducted in obese, non-diabetic cats provide a valuable model for naturally occurring insulin resistance, and the findings reveal clear differences between drug classes. The most compelling evidence comes from the thiazolidinedione (TZD) class, with both darglitazone and pioglitazone significantly improving insulin sensitivity and lipid metabolism [26,28]. However, this presents a significant translational gap, as improvements in surrogate metabolic endpoints in experimental models do not always translate to clinical efficacy in diseased patients. The complete lack of clinical trials evaluating TZDs in the diabetic cats included in this review (Table 2) means their therapeutic role remains a major unanswered question.

Other drug classes showed more nuanced effects. The SGLT2 inhibitor velagliflozin also showed potential; while it did not significantly change the rate of glucose clearance, the cats required less insulin to clear the same amount of glucose. It is plausible that by inducing glucosuria, the drug reduces the overall glucose burden, thereby alleviating glucotoxicity and indirectly improving insulin sensitivity [29]. In contrast, the GLP-1 agonist exenatide did not improve insulin sensitivity in obese cats, although it did reduce food intake and body weight [38]. This lack of effect on insulin sensitivity could suggest that obese cats may have a degree of inherent incretin resistance. Similarly, chromium supplementation had no discernible effect on glucose tolerance [25].

However, when interpreting these findings collectively, several confounding factors must be considered. The studies used different drug formulations and treatment durations, and while all focused on obesity as a model, the baseline degree of obesity and insulin resistance was not uniform across all trials. This inherent heterogeneity makes a direct comparison of the magnitude of the effect between drug classes challenging.

Taken together, these findings suggest that in the insulin-resistant, pre-diabetic feline state, strategies that reduce the glucose load independent of pancreatic insulin secretion (either by enhancing peripheral glucose uptake or promoting urinary excretion) may be more impactful than attempting to modulate insulin secretion itself.

#### 4.3.2. Insights into Pathophysiological Mechanisms

Two studies provided insights into how NIADs interact with the specific pathological processes of FDM, moving beyond simple metabolic measurements. The findings from Hoenig et al. [9] are particularly stark, offering a compelling mechanistic explanation for the poor clinical outcomes of sulfonylureas seen in Table 2. By demonstrating that glipizide-induced hyperinsulinemia accelerates islet amyloid deposition, the study suggests that this drug class may be actively detrimental to the feline pancreas. However, a key limitation when interpreting this finding is the artificial nature of the experimental model, which may not perfectly replicate the slower, progressive pathophysiology of spontaneous FDM.

In a different model targeting a common clinical challenge, the negative finding from Leal et al. [5] is also clinically significant. Their demonstration that metformin was ineffective at preventing insulin resistance following a depot glucocorticoid injection dampens enthusiasm for its prophylactic use. It is important to consider, however, that this model represents an acute, high-dose pharmacological challenge, and these findings may not be generalizable to cats receiving chronic, low-dose oral corticosteroid therapy. Collectively, these experimental studies underscore the limitations and potential harms of older NIADs when examined through a pathophysiological lens.

#### 4.3.3. Pharmacokinetic Considerations

While several pharmacokinetic (PK) studies of NIADs in healthy, normal-weight cats exist in the literature, only one study met the inclusion criteria for this systematic review by specifically evaluating PK in an at-risk population. This is a critical distinction, as the obese state can alter drug absorption and metabolism. The randomized crossover trial by Clark et al. [36] investigated the oral bioavailability of pioglitazone in both lean and obese cats. Their key finding was that oral bioavailability was high (mean 86%) and, importantly, was not significantly different between the two groups. This suggests that, for pioglitazone, obesity does not substantially alter its pharmacokinetics and that dose adjustments based on body condition may not be necessary. This robust bioavailability in an at-risk model provides a strong pharmacological foundation for the positive effects on insulin sensitivity observed in other trials using this drug class [26,28].

However, these valuable PK data must be interpreted within the context of the study’s design. The parameters were determined after a single oral dose, and it remains unknown if chronic administration could alter the drug’s disposition through mechanisms such as enzyme induction. Furthermore, these data were obtained from healthy, euglycemic cats; the presence of clinical FDM mellitus itself could potentially alter drug absorption and metabolism.

#### 4.3.4. Strengths and Limitations of the Evidence

In stark contrast to the clinical studies summarized in Table 2, the evidence from the at-risk and experimental models is characterized by a generally higher level of internal validity. As detailed in Table 4, the majority of these studies (8 of 10) were RCTs, many of which employed robust placebo-controlled and crossover designs [5,9,22,26,28,29,36,37,38]. This rigorous methodology minimizes confounding and allows for more confident conclusions about the direct pharmacological effects of the agents within the specific context of the model studied.

However, the primary limitation of this entire body of evidence is its uncertain external validity, or generalizability to the clinical diabetic population. These studies were conducted in healthy cats (lean or obese) or cats with artificially induced disease states, not in client-owned cats with naturally occurring, often multi-morbid, FDM. For instance, while thiazolidinediones consistently improved insulin sensitivity in obese cats, it is unknown if this effect would be sufficient to overcome the complex pathophysiology of a cat with long-standing clinical diabetes. Similarly, while the findings on islet amyloidosis are mechanistically compelling, they are derived from a non-clinical model of the disease. Therefore, the findings from Group B should be interpreted primarily as valuable mechanistic and hypothesis-generating evidence, rather than as direct proof of clinical efficacy for the treatment of FDM.

### 4.4. Overall Limitations of the Systematic Review

This systematic review has several limitations that should be considered when interpreting its findings. First, while our search strategy was broad and included terms for multiple classes of NIADs, no primary clinical trials meeting our PICO criteria were identified for dipeptidyl peptidase-4 (DPP-4) inhibitors or meglitinides. The potential role of these therapies is discussed in recent reviews [40], but the lack of primary clinical data represents a significant gap in the literature, and consequently, a limitation in the scope of our synthesis.

Second, the potential for publication bias cannot be excluded. It is possible that studies with negative or inconclusive results were less likely to be published, which could lead to an overestimation of the efficacy of some agents included in this review. As discussed previously, the marked heterogeneity across the included studies prevented a quantitative meta-analysis. This was particularly true for the SGLT2 inhibitor subgroup, where pooling was considered but deemed methodologically inappropriate, primarily due to the significant variability in the primary efficacy outcome definitions used across the pivotal trials. As the metrics for “treatment success” were not comparable, combining these disparate outcomes would have been misleading. We therefore proceeded with a narrative synthesis, which relies on qualitative interpretation.

A further limitation is the lack of prospective registration of our review protocol. While the methodology was established a priori, the absence of a public, time-stamped protocol means that any potential deviations from the initial plan cannot be formally audited, which may increase the risk of reporting bias.

Finally, the literature search for this review was concluded on 4 April 2025. Although the authors have monitored the literature during manuscript preparation, it is possible that new evidence may have emerged after this date.

## 5. Conclusions

This systematic review concludes that the therapeutic landscape for FDM is undergoing a paradigm shift. The evidence supporting the use of traditional NIADs, such as sulfonylureas and biguanides, is scarce and of low methodological quality, suggesting limited clinical utility. In contrast, newer agents, particularly SGLT2 inhibitors, have demonstrated considerable efficacy as an oral monotherapy in recent, large clinical trials, in some cases showing non-inferiority to insulin. However, this promising efficacy is balanced by a distinct and serious safety consideration, namely the risk of eDKA, creating a critical new risk–benefit calculus for clinicians and owners, one that is heavily dependent on the owner’s ability and commitment to adhere to strict home monitoring protocols.

The analysis of the current evidence base also reveals several critical gaps, pointing to clear directions for future research. To build a more robust foundation for evidence-based practice, there is a pressing need for high-quality, double-blinded RCTs, not only to confirm the efficacy rates observed in uncontrolled studies but also to conduct head-to-head comparisons between novel agents. Furthermore, future trials must better reflect clinical reality by evaluating the safety and efficacy of NIADs in cats with common comorbidities, such as chronic kidney disease and pancreatitis. A move toward standardized, clinically meaningful primary outcomes, including long-term diabetic remission rates, is also necessary to allow for more meaningful comparisons between studies. Finally, this review identified a significant translational gap; promising drug classes like the thiazolidinediones, which demonstrated strong positive effects in experimental models, warrant investigation in clinical trials of diabetic cats.

## Figures and Tables

**Figure 1 animals-15-02561-f001:**
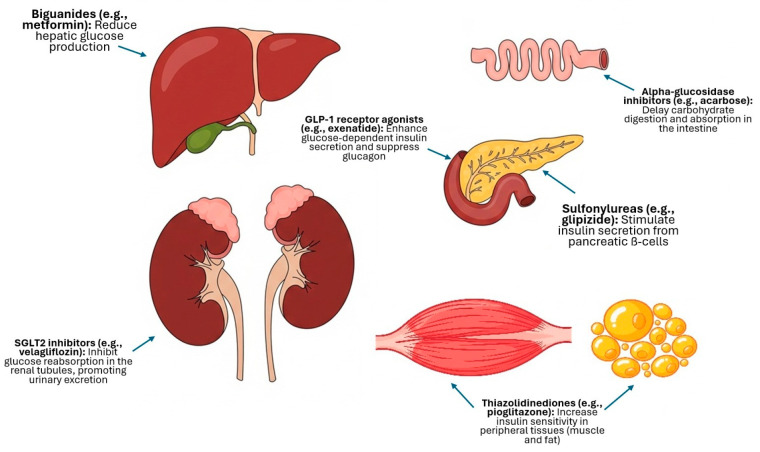
Schematic summary of the primary mechanisms of action of the main classes of non-insulin antidiabetic drugs (NIADs) evaluated in cats.

**Figure 2 animals-15-02561-f002:**
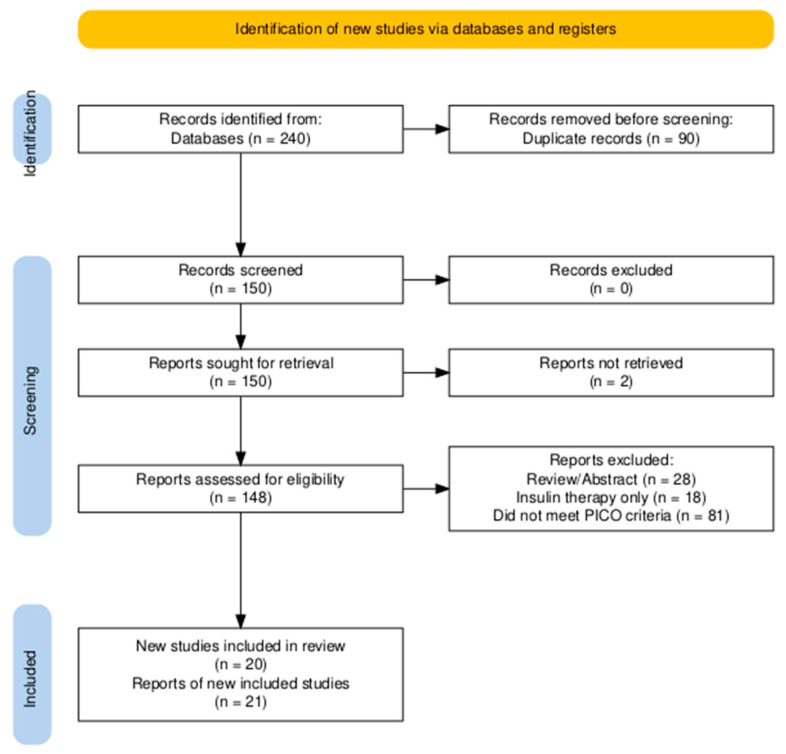
PRISMA flow chart for literature review. Identification of studies via databases. PRISMA = Preferred Reporting Items for Systematic and Meta-analysis.

**Table 1 animals-15-02561-t001:** Search strategy and PICO-based eligibility criteria.

Databases	Google Scholar PubMed Scopus
Date of Search	3 March 2025
Date of Dates included	No restrictions
Search input	Google Scholar: (“diabetes mellitus felina” OR “feline diabetes”) AND (“antidiabéticos orales” OR “oral hypoglycemic agents” OR “biguanidas” OR “sulfonilureas” OR “meglitinidas” OR “inhibidores de alfa-glucosidasa” OR “inhibidores de DPP-4” OR “agonistas de GLP-1” OR “inhibidores de SGLT2” OR “metformin” OR “metformina” OR “glipizide” OR “glipizida” OR “acarbose” OR “acarbosa” OR “sitagliptin” OR “sitagliptina” OR “exenatide” OR “exenatida” OR “dapagliflozin” OR “dapagliflozina” OR “bexagliflozin” OR “velagliflozin”) AND (“clinical trial” OR “ensayo clínico” OR “retrospective study” OR “estudio retrospectivo” OR “case report” OR “reporte de caso”) PubMed: (“Feline Diabetes Mellitus”[MeSH] OR “feline diabetes” OR “diabetes mellitus felina”) AND (“Hypoglycemic Agents”[MeSH] OR “oral hypoglycemic agents” OR “antidiabéticos orales” OR “metformin” OR “metformina” OR “glipizide” OR “glipizida” OR “acarbose” OR “acarbosa” OR “sitagliptin” OR “sitagliptina” OR “exenatide” OR “exenatida” OR “dapagliflozin” OR “dapagliflozina” OR “bexagliflozin” OR “velagliflozin”) AND (“Clinical Trial”[Publication Type] OR “Retrospective Studies”[MeSH] OR “retrospective study” OR “estudio retrospectivo” OR “Case Reports”[Publication Type] OR “case report” OR “reporte de caso”) Scopus: TITLE-ABS-KEY (“feline diabetes” OR “diabetes mellitus felina”) AND TITLE-ABS-KEY (“oral hypoglycemic agents” OR “antidiabéticos orales” OR “biguanides” OR “biguanidas” OR “sulfonylureas” OR “sulfonilureas” OR “meglitinides” OR “meglitinidas” OR “alpha-glucosidase inhibitors” OR “inhibidores de alfa-glucosidasa” OR “DPP-4 inhibitors” OR “inhibidores de DPP-4” OR “GLP-1 agonists” OR “agonistas de GLP-1” OR “SGLT2 inhibitors” OR “inhibidores de SGLT2” OR “metformin” OR “metformina” OR “glipizide” OR “glipizida” OR “acarbose” OR “acarbosa” OR “sitagliptin” OR “sitagliptina” OR “exenatide” OR “exenatida” OR “dapagliflozin” OR “dapagliflozina” OR “bexagliflozin” OR “velagliflozin”) AND TITLE-ABS-KEY (“clinical trial” OR “ensayo clínico” OR “retrospective study” OR “estudio retrospectivo” OR “case report” OR “reporte de caso”)
PICO	PICO A P: Cats with naturally acquired diabetes mellitus (including newly diagnosed and previously treated). I: Treatment with sulfonylureas, alpha-glucosidase inhibitors, biguanides, GLP-1 receptor agonists, or SGLT2 inhibitors. C: Insulin therapy, placebo, or no concurrent control (baseline comparison) O: Glycemic control and adverse effects. PICO B P: Healthy adult cats with risk factors for FDM (eg, obesity), under challenge with diabetogenic drugs (eg, corticosteroids), or in experimental models of FDM. I: Administration of sulfonylureas, thiazolidinediones, biguanides, GLP-1 receptor agonists, SGLT2 inhibitors, or chromium. C: Placebo or an active comparator (insulin). O: Markers of insulin sensitivity, glucose metabolism, pathological changes, and safety.
Inclusion criteria	Examines the efficacy, safety profile, and adverse effects of NIADs used in cats with FDM and cats predisposed to diabetes.
Exclusion criteria	Not relevant to the PICO question. Not peer-reviewed. Not original research. Abstract only. Healthy cats and cats without risk of developing diabetes. Single case reports unrelated to antidiabetics, or review articles not containing primary data.

PICO = Population, Intervention, Comparison and Outcome.

**Table 2 animals-15-02561-t002:** Characteristics of included clinical studies in cats with naturally acquired diabetes mellitus.

Author (Year) [Cite]	Study Design	Population (n, Characteristics)	Intervention (Drug, Dose, Duration)	Comparator	Reported Efficacy Outcomes	Reported Adverse Effects	Risk of Bias
Nelson et al. (1993) [11]	PCUCT	n = 20 cats with naturally acquired FDM (most non-ketotic). Mean age: 9.5 years. The group was generally overweight/obese. History of prior treatment in some cases.	Glipizide: 5 mg/cat, orally, every 12 h. Duration of follow-up was 12 weeks.	None. Cats served as their own controls.	Three response groups identified: 25% (5/20) had good glycemic control. 35% (7/20) failed to respond. 40% (8/20) had a partial response.	Vomiting (3 cats), hypoglycemia (3 cats), and increased serum hepatic enzyme activities (3 cats) were reported.	Moderate
Feldman et al. (1997) [10]	PCUCT	n = 50 cats with recently diagnosed, untreated FDM. Mean age: 10.4 years; mean weight: 5.3 kg. All neutered. Non-ketotic.	Glipizide: 5 mg/cat, orally, every 12 h, administered in treatment phases over 50 weeks.	None. The study included a “no medication” phase as part of its design.	56% (28/50) were treatment failures and switched to insulin. Of the remaining 22, 13 (26% of total) showed clinical/metabolic improvement, and 6 (12% of total) had their FDM resolved.	Transient anorexia and vomiting (8 cats), transient icterus with elevated liver enzymes (4 cats).	Serious
Mazzaferro et al. (2003) [18]	NRCCT	n = 24 client-owned diabetic cats (18 diet/acarbose, 6 diet-only controls). All with a history of obesity. Median age: 10 years.	Acarbose (12.5 mg/cat, orally, BID) combined with a low-carbohydrate diet and concurrent insulin therapy for 4 months.	Low-carbohydrate diet with concurrent insulin therapy.	In the main group (n = 18), 61% (11/18) became ‘responders’ and discontinued insulin. The authors concluded that acarbose likely had a minimal effect compared to the diet alone.	The article does not explicitly report any adverse effects associated with the acarbose treatment.	Serious
Nelson et al. (2004) [14]	PUCCS	Phases 1 and 2: n = 8 healthy adult cats. Phase 3: n = 5 cats with newly diagnosed, naturally acquired diabetes mellitus (Mean age: 11 years. Two cats were obese).	Phases 1 and 2: Metformin dose-finding study in healthy cats. Phase 3: Metformin, orally, with a dose escalation protocol up to 50 mg/cat BID for up to 8 weeks in diabetic cats.	None.	Only 20% (1/5) of cats achieved glycemic control. 60% (3/5) failed to respond and were switched to insulin. 1 cat died unexpectedly.	Lethargy, inappetence, vomiting, and weight loss were noted in healthy cats during the safety phase. One diabetic cat died (cause undetermined, but GI hemorrhage was found).	Serious
Riederer et al. (2016) [19]	NRPCT	n = 30 client-owned cats with newly diagnosed FDM. Median age: ~9.5 years. Treated with insulin and a low- carbohydrate diet.	Exenatide extended release (200 µg/kg), SC, once weekly for up to 16 weeks, as an adjunct to insulin therapy.	Placebo (0.9% saline), SC, once weekly, as an adjunct to insulin therapy.	Primary Outcome: No significant difference was found in the remission rate: 40% (6/15) in the exenatide group vs. 20% (3/15) in the placebo group (*p* = 0.427). Secondary Analysis (Krämer et al., 2020) [21]: Exenatide treatment was found to significantly reduce glycemic variability at weeks 6 and 10 compared to placebo.	Decreased appetite (60% vs. 20%) and vomiting (53% vs. 40%) were more frequent in the exenatide group, but the difference was not statistically significant.	Serious
Scuderi et al. (2018) [2]	RCT	n = 8 client-owned diabetic cats (Body condition score [BCS] ≥ 5/9). Median age: 12 years. All previously treated with insulin glargine for ≥1 month.	Exenatide (short-acting): 1 µg/kg, SC, twice daily for 6 weeks, as an adjunct to insulin therapy.	Placebo (0.9% saline) injection.	Exenatide significantly lowered the required daily insulin dose and induced greater weight loss compared to placebo. 25% (2/8) of cats achieved remission with exenatide vs. 0% with placebo.	Treatment was well tolerated. 2 cats required a temporary dose reduction due to anorexia and weakness associated with hypoglycemia.	Low
Benedict et al. (2022) [31]	PUCCS	n = 5 client-owned cats with poorly regulated FDM on insulin therapy. Median age: 8 years.	Bexagliflozin (10–15 mg/cat), orally, once daily for 4 weeks, as an adjunct to a reduced insulin protocol.	None. Pre-post comparison.	Significant reduction in insulin dose requirement in all cats (*p* = 0.015), with insulin discontinued in 2/5 cats. Significant decrease in mean BG (*p* = 0.022).	No significant adverse effects occurred. No episodes of hypoglycemia were documented.	Serious
Hadd et al. (2023) [32]	POOLHCT	n = 84 client-owned cats with newly diagnosed FDM. Median age: 10.8 years.	Bexagliflozin: 15 mg/cat, orally, once daily for at least 56 days.	None (historically controlled).	84% (68/81 evaluable cats) achieved treatment success (defined as glycemic control and improvement of at least one clinical sign) at day 56	Common: emesis (50%), diarrhea (38%), anorexia (37%). Serious: 8 cats experienced SAEs, including 4 with known or presumed eDKA. 3 deaths/euthanasias occurred.	Serious
Niessen et al. (2024) [30]	ROLACT	n = 127 client-owned diabetic cats (n = 116 for efficacy analysis). Included both newly diagnosed and previously insulin-treated cats. Mean age: 11 years.	Velagliflozin: 1 mg/kg, orally, once daily for up to 91 days.	Insulin (Caninsulin), SC, twice daily, with dose adjusted by clinicians.	The study demonstrated non-inferiority. At Day 45, 54% (29/54) of velagliflozin-treated cats were treatment successes, compared to 42% (26/62) of insulin-treated cats.	Adverse effects differed by group. Velagliflozin group: most frequent were diarrhea (38%) and positive urine culture (31%); eDKA occurred in 4/61 cats. Insulin group: most frequent was hypoglycemia (clinical & non-clinical), occurring in 53% of cats.	Some
Behrend et al. (2024) [1]	POLBCT	n = 252 client-owned diabetic cats. Included both newly diagnosed (85%) and previously insulin-treated (15%) cats. Median age: 11 years.	Velagliflozin oral solution: 1 mg/kg, orally, once daily for up to 180 days.	None (baseline-controlled).	At day 180, 81% of remaining cats had BG and/or fructosamine within reference ranges. 88.6% and 87.7% showed owner-reported improvement in polyuria and polydipsia, respectively.	Ketoacidosis occurred in 7.1% of cats, with most cases being euglycemic. Ketonuria without acidosis occurred in an additional 6.7%. Most episodes occurred within the first 14 days of treatment.	Serious

PCUCT: Prospective Clinical Trial (Uncontrolled). NRCCT: Non-Randomized, Controlled Clinical Trial. PUCCS: Prospective, Uncontrolled Clinical Trial (Case Series). NRPCT: Non-Randomized, Placebo-Controlled Clinical Trial. RCT: Randomized, Double-Blinded, Placebo-Controlled, Crossover Trial. POOLHCT: Prospective, Open-Label, Historically Controlled Clinical Trial. ROLACT: Randomized, Open-Label, Active-Controlled, Non-Inferiority Trial. POLBCT: Prospective, Open-Label, Baseline-Controlled Clinical Trial.

**Table 3 animals-15-02561-t003:** Reported breed composition of cats in the included clinical studies.

Author (Year) [Cite]	Reported Breed Composition
Nelson et al. (1993) [11]	Not specified
Feldman et al. (1997) [10]	Not specified; described as cats from a referral population
Mazzaferro et al. (2003) [16]	Not specified
Nelson et al. (2004) [14]	Not specified
Riederer et al. (2016) [18]	Primarily Domestic Shorthair and Longhair
Scuderi et al. (2018) [2]	Not specified
Benedict et al. (2022) [31]	Domestic Shorthair (n = 3), Domestic Longhair (n = 2)
Hadd et al. (2023) [32]	Primarily domestic mixed-breed cats from North America
Niessen et al. (2024) [30]	Primarily Domestic Shorthair and Longhair
Behrend et al. (2024) [1]	Primarily Domestic Shorthair/Longhair; purebred cats < 10%

**Table 4 animals-15-02561-t004:** Characteristics of Included Pre-clinical Studies in At-Risk and Experimental Feline Models.

Author (Year) [Cite]	Study Design	Population (n, Characteristics)	Intervention (Drug, Dose, Duration)	Comparator	Reported Efficacy Outcomes	Reported Adverse Effects	Risk of Bias
Cohn et al. (1999) [25]	NRPCT	n = 19 healthy, non-diabetic cats. Divided into 3 groups: non-obese placebo (n = 6), non-obese Cr (n = 6), and obese Cr (n = 7).	Chromium picolinate: 100 µg/cat, orally, every 24 h for 6 weeks.	Placebo (calcium phosphate).	No significant effect. Chromium supplementation did not alter responses to IV Glucose Tolerance Testing (IVGTT) in either the non-obese or obese groups.	No adverse health effects were observed. A significant decrease in serum potassium was noted in obese cats, but values remained within the reference range.	Moderate
Hoenig et al. (2000) [9]	RCT-EM	n = 8 healthy male cats, in FDM was experimentally induced via partial pancreatectomy and hormonal treatment.	Glipizide: 5 mg/cat, orally, 2–3 times daily for 18 months.	Insulin (Humulin N).	The primary outcome was islet amyloid formation. 100% (4/4) of glipizide-treated cats developed islet amyloid, while only 25% (1/4) of insulin-treated cats did.	No clinical adverse effects were reported. The study was terminal.	Some concerns
Hoenig & Ferguson (2003) [26]	RCT-PC	n = 22 healthy, non-diabetic adult female cats (4 lean, 18 obese). Obese cats randomized to placebo (n = 9) or darglitazone (n = 9).	Darglitazone: 2 mg/kg, orally, once daily for 42 days.	Placebo capsule.	Darglitazone significantly improved insulin sensitivity (reduced area under the curve (AUC) for glucose and insulin in IVGTT) compared to placebo. Also significantly lowered cholesterol and triglyceride concentrations.	The drug was well tolerated. No negative clinical effects were reported.	Some concerns
Clark et al. (2012) [36]	RCT-PK	n = 12 healthy adult cats, divided into lean (n = 6) and obese (n = 6) groups.	Pioglitazone: single oral dose of 3 mg/kg.	IV administration of the same drug (bioavailability study).	This was a pharmacokinetic study. It found that the oral bioavailability of pioglitazone was high (mean 86%) and not significantly different between lean and obese cats.	No adverse effects were observed during the study.	Some concerns
Clark et al. (2014) [28]	RCT-3WC	n = 12 obese, healthy, non-diabetic adult cats (6 male, 6 female). Age: 5–7 years.	Pioglitazone: 1 mg/kg or 3 mg/kg, orally, once daily for 7-week periods.	Placebo capsule.	The 3 mg/kg dose significantly improved insulin sensitivity and lowered serum triglyceride and cholesterol concentrations compared to placebo.	No adverse effects attributable to pioglitazone were evident. One cat died during sedation, considered unlikely to be caused by the drug.	Some concerns
Hall et al. (2015) [22]	RCT-PCCT	n = 8 healthy adult cats (6 male, 2 female). Three were lean (BCS 5/9) and five were overweight (BCS 6–8/9).	Liraglutide: single SC injection of 3 or 6 nmol/kg.	Placebo (saline) injection.	Liraglutide significantly increased glucose-stimulated insulin secretion and suppressed glucagon secretion compared to placebo. It did not have a significant effect on food intake.	The study did not report any adverse effects.	Some concerns
Rudinsky et al. (2015) [37]	RCT-PCCT	n = 6 healthy, lean, adult n = 6 healthy, adult neutered male cats. Three were classified as lean (BCS 5/9) and three as overweight (BCS 6–7/9) male cats.	Exenatide ER: single SC injection of 2 mg/cat.	Placebo (vehicle) injection.	Did not significantly alter glucose or insulin response during an IVGTT. Induced a significant reduction in food intake on days 2–4 post-injection.	Vomiting was observed in 5 of 6 cats within 24 h of exenatide injection.	Some concerns
Hoelmkjaer et al. (2016) [38]	RCT-DBPC	n = 11 obese, but otherwise healthy, client-owned cats.	Exenatide extended release: 2 mg/cat, SC, once weekly for 8 weeks.	Placebo injection.	Exenatide significantly decreased food intake and led to weight loss compared to placebo. It did not significantly improve insulin secretion or insulin sensitivity during an IVGTT.	Transient and mild hyporexia and vomiting were the most common adverse effects observed.	Some concerns
Hoenig et al. (2018) [29]	NRPCT	n = 12 obese, healthy, non-diabetic adult cats (6 male, 6 female). Median age: 6 years.	Velagliflozin: 1 mg/kg, orally, once daily for 35 days.	Placebo capsule.	Velagliflozin significantly increased urinary glucose excretion but did not significantly alter overall glucose tolerance. It did lower the insulin response, suggesting improved insulin sensitivity.	“All cats tolerated treatment well.” Soft stool was observed in 2 cats in each group.	Moderate
Leal et al. (2022) [5]	RCT	n = 28 client-owned, non-diabetic cats receiving a single dose of methylprednisolone acetate (MPA). Mean age: 5.6 years.	Two intervention groups: 1) Metformin (25 mg/cat, orally, once daily) or 2) A commercial obesity/diabetes (O and D) diet. Both for 30 days.	Control group (n = 10) that received only the MPA injection.	No significant protective effect. Neither metformin nor the O and D diet was effective in preventing the insulin resistance induced by the MPA injection.	No side effects associated with metformin or the diet were reported by the owners.	High

NRPCT: Non-Randomized, Placebo-Controlled Clinical Trial. RCT-EM: Randomized Controlled Trial, Experimental Model. RCT-PC: Randomized, Placebo-Controlled Trial. RCT-PK: Randomized Crossover Trial, Pharmacokinetic Study. RCT-3WC: Randomized, Placebo-Controlled, 3-way Crossover Trial. RCT-PCCT: Randomized, Placebo-Controlled, Crossover Trial. RCT-DBPC: Randomized, Double-Blinded, Placebo-Controlled, Crossover Trial. RCT: Randomized Clinical Trial.

## Data Availability

All data generated or analyzed during this study are included in this published article.

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
