# Peer review of "Efficacy and Safety of Non-Insulin Antidiabetic Drugs in Cats: A Systematic Review"

_animals, 2025, doi:10.3390/ani15172561_

Round 1
Reviewer 1 Report
Comments and Suggestions for Authors
Indeed, despite the clinical significance of feline diabetes, the literature lacks a comprehensive, methodologically robust review summarizing current treatment options for diabetic cats. Particularly noteworthy is the emerging interest in non-insulin antidiabetic drugs (NIADs), which are gaining popularity among veterinary clinicians in recent years.
The authors have conducted a systematic review of 20 studies, covering various classes of drugs (sulfonylureas, biguanides, SGLT2 inhibitors, GLP-1 receptor agonists, etc.). The analysis is thorough and effectively distinguishes between clinical studies (in diabetic cats) and experimental risk models.
The review was conducted in accordance with PRISMA guidelines, including a detailed description of the database search strategies.
The authors critically appraise both the strengths and weaknesses of the existing evidence, as well as the limitations of their own review. They emphasize issues such as small sample sizes, heterogeneity of results, and the absence of randomized controlled trials (RCTs) for several drug classes. Importantly, they highlight the risk of euglycemic diabetic ketoacidosis (eDKA) as a significant adverse effect associated with SGLT2 inhibitors.
Suggestions for Improvement:
- The lack of a quantitative meta-analysis is a serious shortcoming. Despite the substantial amount of available data, no meta-analysis was performed. While the heterogeneity of the included studies provides some justification, the authors could have employed methods that allow for subgroup meta-analyses (e.g., focusing solely on SGLT2 inhibitors). In some sections, there is a tendency to extrapolate results narratively without support from robust statistical analyses.
- I recommend providing a detailed summary table of study populations from the included articles, explicitly stating the breed composition of the enrolled cats. Although the authors mention the predisposition of Burmese cats to diabetes, the analysis does not clarify whether and how breed distribution was accounted for, which limits the generalizability of the findings.
- While the risk of eDKA is acknowledged, the authors occasionally seem to overstate the safety profile of SGLT2 inhibitors in clinical practice. For instance, in the conclusions, they somewhat underplay the practical challenges, such as the risk of owner unawareness regarding eDKA and the demands of appropriate at-home monitoring.
Author Response
Author's Reply to Reviewer 1
We thank the reviewer for their positive evaluation and for the valuable suggestions to improve our manuscript. We have carefully considered all comments and have revised the manuscript accordingly. Below, we provide a point-by-point response. The locations of the changes in the revised manuscript are indicated, and new or modified text is quoted for clarity.
Comment 1: The lack of a quantitative meta-analysis is a serious shortcoming. Despite the substantial amount of available data, no meta-analysis was performed. While the heterogeneity of the included studies provides some justification, the authors could have employed methods that allow for subgroup meta-analyses (e.g., focusing solely on SGLT2 inhibitors). In some sections, there is a tendency to extrapolate results narratively without support from robust statistical analyses
Response 1: We thank the reviewer for highlighting the importance of quantitative analysis and for this insightful suggestion. We carefully considered the possibility of conducting a meta-analysis, particularly for the SGLT2 inhibitor subgroup, as the reviewer rightly suggests.
However, after a detailed methodological evaluation, we maintained our decision not to proceed due to a critical level of heterogeneity that extended beyond just the study designs. The primary obstacle was the marked variability in the definition of the primary efficacy outcomes used across the pivotal trials. As the metrics and thresholds for ‘treatment success’ were not comparable (e.g., composite scores vs. glycemic parameters), statistically pooling these disparate outcomes into a single summary estimate would have been methodologically inappropriate and could have produced a misleading conclusion, a point also emphasized in the PRISMA 2020 guidelines [34].
To address the reviewer's concern, we have expanded upon this justification within our limitations section. This change can be found in the revised manuscript in Section 4.4, "Overall Limitations of the Systematic Review" [Page 18, Paragraph 2]. The updated text reads as follows:
"As discussed previously, the marked heterogeneity across the included studies prevented a quantitative meta-analysis. This was particularly true for the SGLT2 inhibitor subgroup, where pooling was considered but deemed methodologically inappropriate, primarily due to the significant variability in the primary efficacy outcome definitions used across the pivotal trials. As the metrics for "treatment success" were not comparable, combining these disparate outcomes would have been misleading. We therefore proceeded with a narrative synthesis, which relies on qualitative interpretation."
Comment 2: I recommend providing a detailed summary table of study populations from the included articles, explicitly stating the breed composition of the enrolled cats. Although the authors mention the predisposition of Burmese cats to diabetes, the analysis does not clarify whether and how breed distribution was accounted for, which limits the generalizability of the findings.
Response 2: We thank the reviewer for this excellent suggestion, which enhances the transparency and the discussion on the generalizability of our findings. We agree completely.
To address this point, we have made two changes to the manuscript. First, we added a sentence to refer to the new table. This change can be found in Section 3.2.1, "Characteristics of Included Studies" [Page 6, Paragraph 1, last line]:
"The reported breed composition of the cats included in these trials is summarized in Table 2."
Second, we have added a new summary table (Table 3) to the manuscript. This table can be found following Table 2 [Page 8]. The new table is as follows:
Table 3. Reported breed composition of cats in the included clinical studies.
|
Author (Year) [cite] |
Reported Breed Composition |
|
Nelson et al. (1993) [10] |
Not specified |
|
Feldman et al. (1997) [9] |
Not specified; described as cats from a referral population |
|
Mazzaferro et al. (2003) [16] |
Not specified |
|
Nelson et al. (2004) [13] |
Not specified |
|
Riederer et al. (2016) [18] |
Primarily Domestic Shorthair and Longhair |
|
Scuderi et al. (2018) [2] |
Not specified |
|
Benedict et al. (2022) [30] |
Domestic Shorthair (n=3), Domestic Longhair (n=2) |
|
Hadd et al. (2023) [31] |
Primarily domestic mixed-breed cats from North America |
|
Niessen et al. (2024) [29] |
Primarily Domestic Shorthair and Longhair |
|
Behrend et al. (2024) [1] |
Primarily Domestic Shorthair/Longhair; purebred cats <10% |
This addition makes it clear that the vast majority of evidence has been generated in mixed-breed domestic cats, which reinforces our discussion point that robust evidence in specific predisposed purebred cats is currently lacking.
Comment 3: While the risk of eDKA is acknowledged, the authors occasionally seem to overstate the safety profile of SGLT2 inhibitors in clinical practice. For instance, in the conclusions, they somewhat underplay the practical challenges, such as the risk of owner unawareness regarding eDKA and the demands of appropriate at-home monitoring.
Response 3: We thank the reviewer for this important observation regarding the clinical interpretation and practical implications of our findings. We agree that the practical challenges associated with eDKA monitoring are a critical component of the risk-benefit discussion and that our conclusions should more strongly reflect this. The reviewer's point that our conclusion was not fully supported by the safety data due to this underemphasis is well-taken.
To address this, we have revised the language in both the Discussion and the Conclusion sections to place greater emphasis on the significant practical demands placed on cat owners and the crucial role of their vigilance and education. The updated texts read as follows:
In Section 4.2.3, "Safety Profile and Adverse Events" [Page 15, Paragraph 3, last sentence], the final sentence now reads:
"Ultimately, this highlights a fundamental risk-benefit trade-off: the high efficacy reported for SGLT2 inhibitors must be weighed against the significant risk of eDKA, a life-threatening complication that requires a high degree of owner commitment and education, posing a significant practical challenge for both veterinarians and cat owners in a clinical setting."
In Section 5, "Conclusions" [Page 18, Paragraph 1], the relevant sentence now reads:
"However, this promising efficacy is balanced by a distinct and serious safety consideration, namely the risk of eDKA, creating a critical new risk-benefit calculus for clinicians and owners, one that is heavily dependent on the owner’s ability and commitment to adhere to strict home monitoring protocols."
Once again, we thank the reviewers for their time and valuable contributions, which we believe have significantly strengthened our manuscript. We hope that the revised version is now suitable for publication in Animals
Reviewer 2 Report
Comments and Suggestions for Authors
The review is original and could be very relevant for comparative medicine in the field of Diabetes mellitus.
This review shows that veterinarians and cat owners now have a viable oral alternative to insulin, but the choice of treatment involves a new risk-benefit discussion centered on owner education and vigilance for signs of euglycemic diabetic ketoacidosis.
The methodology of the study is very good. The systematic review was conducted following PRISMA guidelines. Major databases were searched for studies evaluating NIADs in diabetic cats or at-risk/experimental models.
The results could be very important for veterinarians and owners of diabetic cats since authors findings demonstrated that traditional agents (glipizide, metformin, acarbose) showed limited efficacy in diabetes mellitus. Newer SGLT2 inhibitors (bexagliflozin, velagliflozin) demonstrated high treatment success rates and non-inferiority to insulin, but were associated with a significant risk of euglycemic diabetic ketoacidosis.
The conclusions are relevant for authors findings.
I suggest some minor corrections.
- For a better readability authors should include at least a schematic Figure about mechanism of action of different antidiabetic drugs.
- Introduction and Discussions could be improved, including actual data about failure of insuline therapy in cats with non-insulin dependent diabetes mellitus
For Introduction and Discussions, improvement authors may see also
a.Screening diabetic cats for hypersomatotropism: performance of an enzyme-linked immunosorbent assay for insulin-like growth factor 1, Journal of Feline Medicine and Surgery 2014 16: 82-88, DOI: 10.1177/1098612X13496246
b.Madalina Rosca, Mihai Musteata, Carmen Solcan, Gabriela Dumitrita Stanciu, Gheorghe Solcan, Feline Hypersomatotropism, an Important Cause for the Failure of Insulin Therapy, Bulletin UASVM Veterinary Medicine 71(2) / 2014, 298-304, DOI:10.15835/buasvmcn-vm: 10288
3 Minor Editing corrections
-Authors citation should be made according to MDPI Guide for authors, eg Author name et al (reference nr, not year)
Tables should be numbered from 1 to n (not A,B..)
Author Response
Author's Reply to Reviewer 2
We thank the reviewer for their positive evaluation and for the valuable suggestions to improve our manuscript. We have carefully considered all comments and have revised the manuscript accordingly. Below, we provide a point-by-point response. The locations of the changes in the revised manuscript are indicated, and new or modified text is quoted for clarity.
Comment 1: For a better readability authors should include at least a schematic Figure about mechanism of action of different antidiabetic drugs
Response 1: We thank the reviewer for this excellent suggestion to improve the manuscript's readability. We agree that a schematic figure illustrating the mechanisms of action of the different drug classes would be a valuable addition for the reader.
To address this, we have made two changes to the manuscript. First, we have now created and inserted a new figure (now Figure 1) [Page 3]. Second, we have added a reference to this new figure in the Introduction, Section 1 [Page 2, Paragraph 3, last sentence]. The updated text reads as follows:
"The distinct mechanisms of action for these drug classes are summarized in Figure 1."
The new Figure 1, can be found on [Page 3] of the revised manuscript.
Comment 2: Introduction and Discussions could be improved, including actual data about failure of insuline therapy in cats with non-insulin dependent diabetes mellitus. For Introduction and Discussions, improvement authors may see also
a.Screening diabetic cats for hypersomatotropism: performance of an enzyme-linked immunosorbent assay for insulin-like growth factor 1, Journal of Feline Medicine and Surgery 2014 16: 82-88, DOI: 10.1177/1098612X13496246
b.Madalina Rosca, Mihai Musteata, Carmen Solcan, Gabriela Dumitrita Stanciu, Gheorghe Solcan, Feline Hypersomatotropism, an Important Cause for the Failure of Insulin Therapy, Bulletin UASVM Veterinary Medicine 71(2) / 2014, 298-304, DOI:10.15835/buasvmcn-vm:10288
Response 2: The reviewer makes a very important point about contextualizing the need for NIADs by discussing the challenges associated with conventional insulin therapy. We thank the reviewer for this insightful comment and for the useful references provided.
To address this, we have made two changes to the manuscript. First, we have added a sentence to the Introduction (Section 1, Page 2, Paragraph 2) that discusses the challenges of insulin therapy. The new text is as follows:
"Furthermore, achieving adequate glycemic control with insulin can be difficult, with a significant proportion of cats failing to achieve diabetic remission, often due to underlying insulin-resistant conditions such as hypersomatotropism [7]."
Second, we have expanded the Discussion (Section 4.2.2.4, "Comorbidities and Concomitant Medications") [Page 15, Paragraph 1, last sentence]. To acknowledge severe insulin-resistant states, such as acromegaly. The new text in this section reads:
"This is particularly relevant for common comorbidities that cause severe insulin resistance and subsequent treatment failure, such as hypersomatotropism (acromegaly), which may affect up to 25% of diabetic cats and is a leading cause of poor glycemic control [7, 40]."
Comment 3: -Authors citation should be made according to MDPI Guide for authors, eg Author name et al (reference nr, not year) -Tables should be numbered from 1 to n (not A,B..)
Response 3: We thank the reviewer for pointing out these important formatting inconsistencies. We have now revised the entire manuscript to address these points:
- All in-text citations have been changed to the Author et al. [X] format, removing the year of publication from the main text where it was not grammatically essential. This has been applied throughout the manuscript.
- The tables have been renumbered sequentially according to their order of appearance. The original Table A is now Table 2, the newly added breed composition table is Table 3, and the original Table B is now Table 4. All in-text references to these tables have been updated accordingly.
Once again, we thank the reviewers for their time and valuable contributions, which we believe have significantly strengthened our manuscript. We hope that the revised version is now suitable for publication in Animals.
Round 2
Reviewer 1 Report
Comments and Suggestions for Authors
I would like to sincerely thank you for your valuable corrections.